# How to Manage Advanced Differentiated Thyroid Cancer: Step-by-Step Analysis from Two Italian Tertiary Referral Centers

**DOI:** 10.3390/jcm13030708

**Published:** 2024-01-25

**Authors:** Paola Vincenza Sartori, Sara Andreani, Loredana De Pasquale, Iuliana Pauna, Antonio Mario Bulfamante, Paolo Salvatore Lorenzo Aiello, Rossella Melcarne, Laura Giacomelli, Marco Boniardi

**Affiliations:** 1General Surgical Department, ASST Brianza-Pio XI Hospital, 20832 Desio, Italy; 2Endocrine Surgery Unit, Niguarda Hospital, 20162 Milan, Italy; sara.andreani@ospedaleniguarda.it (S.A.); iuliana.pauna@ospedaleniguarda.it (I.P.); paolo.aiello@ospedaleniguarda.it (P.S.L.A.); marco.boniardi@ospedaleniguarda.it (M.B.); 3Thyroid and Parathyroid Surgery Service-Otolaryngology Unit, ASST Santi Paolo e Carlo, Department of Health Sciences, University of Milan, 20122 Milan, Italy; loredana.depasquale@asst-santipaolocarlo.it; 4Pediatric Otolaryngology Unit, ASST Fatebenefratelli-Sacco, Buzzi Children Hospital, 20162 Milan, Italy; antonio.bulfamante90@gmail.com; 5Department of Translational and Precision Medicine, Sapienza University of Rome, AOU Umberto I, 00185 Rome, Italy; rossella.melcarne@uniroma1.it; 6Department of General and Specialty Surgery, Sapienza University of Rome, AOU Umberto I, 00185 Rome, Italy; laura.giacomelli@uniroma1.it

**Keywords:** thyroid surgery, cancer, treatment

## Abstract

Background: Differentiated thyroid carcinoma (DTC) has an excellent prognosis; however, advanced disease is associated with a worse prognosis and is relatively common. Surgery followed by RAI treatment remains the mainstream treatment for a large majority of patients with high- and intermediate-risk DTC, but its benefits should be carefully weighed against the potential for harm. The aim of this paper is to critically review the experience in treating advanced DTC at two tertiary referral centers in Italy. Methods: Retrospective analysis of 300 patients who underwent surgery for ADTC over 30 years. Results: The complication rate was 50.33%. A total of 135 patients (45%) remained at regular follow-up, 118 (87.4%) were alive, while 17 (12.6%) were deceased. The mean overall survival at 12 years was 84.8% with a mean of 238 months. Eleven patients (8.1%) experienced a relapse after a median of 13 months. Conclusions: ADTC patients adequately treated can achieve prolonged survival even in the case of metastasis or disease relapse. Patients with ADTC should be referred to high-volume centers with the availability of an extended multidisciplinary team to receive tailored treatment.

## 1. Introduction

The incidence of thyroid cancer continues to rise, in large part due to the detection of incidental small papillary thyroid cancer using high-resolution imaging [1]. Differentiated thyroid carcinoma (DTC) arises from thyroid follicular cells and is the most prevalent endocrine malignant tumor.

Macroscopically, it may present as a solitary or multifocal lesion with an average size of 2–3 cm. The lesion may infiltrate the capsule and have poorly demarcated margins; in addition, areas of fibrosis, cystic areas, and calcifications may be present.

Microscopically, a papillary thyroid carcinoma (PTC) has papillae containing a fibrovascular core in the center; these papillae are covered a with mono- or multilayered cuboid epithelium.

Nuclear atypicalities are the most diagnostically important feature: they include ground-glass or Orphan Annie’s eye nucleus, nuclear pseudoinclusions, due to invaginations of the cytoplasm that are visible in a transverse section, and nuclear incisions.

The presence of concentric calcifications (psammomatous bodies) is often observed and may strengthen the diagnostic suspicion.

Several histologic variants of PTC have been described, with different prognostic impact.

Papillary thyroid carcinoma (PTC), follicular thyroid carcinoma (FTC), and Hürthle cell carcinoma (HTC) are the most common variants of DTC representing more than 90% of all thyroid cancers.

A PTC is called a papillary microcarcinoma (PTMC) if the size of the lesion is less than 1 cm. The incidence of such tumors has been growing rapidly in recent decades and has a favorable prognosis.

PTCs tend to spread mainly by the lymphatic route; on the contrary, follicular thyroid carcinoma metastasizes by hematogenous spread [2,3].

The most frequent sites of metastasis are lung and bone; more rarely, it may affect other regions such as the central nervous system, subcutis, and liver [4,5].

Risk factors known to be strongly associated with DTC include a history of DTC in a first-degree relative, exposure to radiation during childhood, and rare genetic syndromes [6]. The gender of women, a history of goiter or thyroid nodule, a family history of thyroid cancer, a low-iodine diet, radiation exposure, and obesity are other possible risk factors involved in the development of thyroid cancer [7].

The association between Hashimoto’s thyroiditis and papillary thyroid carcinoma is highly controversial [8,9].

The pathogenic hypotheses underlying this association are (1) a prolonged TSH increase over time, which, by a trophic effect on follicular cells, may lead to an increased risk of tumor pathology; (2) the known correlation that exists between a chronic inflammatory state and carcinoma development [10,11].

Cytologic examination on needle aspiration is the gold standard for the diagnosis of thyroid carcinoma, being a simple, well-tolerated, minimally invasive method with a sensitivity of 97% and specificity of 51% [12].

Indications for needle aspiration depend on the size and risk of malignancy of the nodule after initial evaluation (semeiotic, ultrasonographic, and hematochemical): recent ATA guidelines suggest that needle aspiration should be performed on high- and intermediate-suspicion nodules of malignancy ≥1 cm in size, on low-suspicion nodules ≥1.5 cm in size, and on very-low-suspicion nodules ≥2 cm in size [13].

The examination result is reported according to the “Bethesda System for Reporting Thyroid Cytopathology” classification, comprising six categories: (i) nondiagnostic; (ii) benign; (iii) atypia of undetermined significance; (iv) follicular neoplasm; (v) suspicious for malignancy; and (vi) malignant [12].

Although most differentiated thyroid carcinomas (DTCs) have an excellent prognosis, locally invasive disease portends a worse prognosis and is a relatively common event, representing 13% to 15% of DTC cases. The surgical management of advanced thyroid cancer is complex and challenging. Although surgery represents the mainstay of initial treatment, it may be a source of complications and, if extended to other structures for oncological safety, it may have a negative impact over the quality of life; therefore, its benefits must be carefully weighed against the potential for harm [14]. Surgery followed by radioactive iodine (RAI) treatment to eradicate microscopical disease or distant metastases remains the mainstream treatment for the large majority of patients with high- and intermediate-risk DTC [6]. Furthermore, multikinase inhibitors (MKIs), including sorafenib and lenvatinib, are increasingly being used as alternative medical treatments in the most advanced forms, especially in differentiated thyroid cancer refractory to RAI. In the literature, advanced DTC (ADTC) is defined by the presence of one or more of the following features: local invasion, bulky cervical nodes, and/or distant metastases [15].

Up to now, there are no clear and specific guidelines about the treatment of these tumors, but there is comprehensive agreement on the need for a customized and multidisciplinary approach. 

In recent decades, much emphasis has been placed on the molecular profiling of nodules in order to better characterize and diagnose malignant nodules. The first gene to be evaluated for this purpose is BRAF: the BRAF V600E mutation is present in 28–83% of PTCs depending on the case series [16,17].

According to a 2014 study by Hyeon et al., the specificity and sensitivity of BRAF for detecting carcinomas in nodules classified as atypia of undetermined significance (AUS) is 95% and 70% [18].

Some authors have proposed a larger molecular study by applying a panel of mutations such as BRAF, N-, H-, K-RAS, RET/PTC1, RET/PTC3, and PAX8/PPAR, finding a higher sensitivity than for mutation in the BRAF gene alone without a significant decrease in specificity [19].

Recent studies have developed a genetic risk profile using available molecular tests, providing accurate and robust risk stratification for distant metastasis in patients with DTC. The availability of a preoperative prognosis based on such data could allow more personalized treatment for patients with DTC [20].

However, at present, the use of molecular biology as an ancillary technique is not routinely enrolled in clinical practice because, to date, there is not enough scientific evidence.

The aim of this paper is to critically review the experience of treating advanced DTC in two tertiary referral centers in Milan, Italy.

## 2. Materials and Methods

Retrospective analysis of medical and electronic charts of 300 patients operated for advanced differentiated thyroid cancer (ADTC) at 2 referral centers in Milan, Italy, between May 1992 and September 2022. 

Patients were considered to have advanced disease if they had at least one of the following characteristics: nodal invasion, involvement of adjacent structures, or distant metastases. In total, 22 patients had an invasion of adjacent structures (12 recurrent nerves, 6 infiltration of trachea, 2 infiltration of the larynx one of the esophagus, and 1 subcutaneous tissues). A total of 18 patients were metastatic at diagnosis (16 lung, 1 bone, and 1 liver).

Details about pathology are shown in Table 1.

An exception was granted by the Institutional Review Board evaluation due to the retrospective nature of the study.

Preoperative assessment consisted of neck ultrasound with fine needle biopsy and blood tests (TSH, calcium, PTH, 25—OH vitamin D, and calcitonin blood levels).

All patients underwent a preoperative vocal cords evaluation by video laryngoscopy. All ADTC cases were discussed preoperatively and postoperatively by a multidisciplinary team (MDT) consisting of an endocrinologist, an endocrine surgeon, an otolaryngologist, a pathologist, a radiologist, and a nuclear medicine physician.

All operations were performed by an experienced endocrine surgeon who followed a standard technique.

The surgical procedures were classified as follows: total thyroidectomy (total extracapsular thyroidectomy with standard Kocher incision and high extracapsular dissection of the thyroid gland);extended thyroidectomy (total thyroidectomy plus resection of adjacent structures involved by tumor such as larynx, trachea, esophagus, recurrent laryngeal nerve, prevertebral fascia, carotid vessels);unilateral central compartment node dissection (lymphectomy of Compartments 6 and 7 on the same side of the tumor);bilateral central compartment node dissection (lymphectomy of Compartments 6 and 7 on both sides);unilateral modified cervical neck dissection (neck dissection of Compartments 2-3-4-5 on the same side of the tumor sparing spinal accessory nerve and internal jugular vein);bilateral modified cervical neck dissection (neck dissection of Compartments 2-3-4-5 on both sides, sparing spinal accessory nerve and internal jugular vein).

Recurrent laryngeal nerves were always visualized on both sides and the branches of the superior and inferior thyroid arteries were divided close to the thyroid capsule.

The parathyroid glands were dissected meticulously from the thyroid, and an effort was made to identify all four parathyroids and preserve as many as possible in situ, unless they were encased by the tumor. Any inadvertently removed parathyroid tissue and any gland not involved by cancer that could not be preserved were reimplanted into the sternomastoid muscle.

Since 2018, continuous or intermittent intraoperative neuromonitoring has been routinely used to identify and preserve the recurrent laryngeal nerve.

Hemostasis was routinely achieved by surgical ligations and unipolar or bipolar energy devices; advanced energy devices (either Ultracision, Ligasure, or Thundertbeat) were used in 139 patients.

At least one suction drain was used in all cases, and it was removed when the daily output was less than 15 cc.

After treatment, all patients received levothyroxine therapy at a TSH-suppressive dose.

In total, 3 patients with recurrent disease were treated with new medical drugs (2 Lenvatinib and 1 thyroid kinase inhibitor).

Follow-up was carried out by means of office visits, TSH and thyroglobulin dosage, and neck ultrasound every 6 months for the first year and then once a year.

Up-to-date follow-up data were available for 135 patients, and those with incomplete data were considered to be lost at follow-up. 

Surgical complications were classified and defined as follows: transient hypoparathyroidism (hypocalcemia lasting less than six months)permanent hypoparathyroidism (hypocalcemia lasting more than six months)transient recurrent laryngeal nerve palsy (vocal cord function recovered within 6 months)permanent recurrent laryngeal nerve palsy (vocal cord not functioning after more than 6 months)permanent spinal accessory nerve palsyother complications.

Statistical analysis was performed using IBM SPSS Statistics for Windows Version 27.0 (IBM Corp., Armonk, NY, USA).

Continuous data were compared by a *t*-test, categorical data were compared by the chi-square test, and survival analysis was performed by the Kaplan–Meier survival curve method.

The results are expressed as the mean ± standard deviation.

The significance threshold was set at *p* < 0.05.

## 3. Results

There were 109 males (36.3%) and 191 females with a mean age of 46 ± 14.9 years. 

A total of 257 patients were submitted a total thyroidectomy associated lymph node dissection of variable extent (165 central compartment plus unilateral cervical lymphectomy, 79 central compartment, 13 central compartment plus bilateral cervical lymphectomy, 28 bilateral cervical lymphectomy, 11 unilateral central compartment lymphectomy), while only 2 patients received a thyroid lobectomy because of contralateral previous surgery for benign pathology.

Among the patients submitted an extended thyroidectomy, 8 had recurrent nerve resection, 4 had nerve shaving, 1 patient underwent total laryngectomy, and 6 underwent tracheal shaving, and 1 had esophageal shaving. 

The details of the surgical procedures and complications are reported in Table 2. 

In total, 274 patients had radical surgery (R0), 10 had microscopic residual disease (R1), and in 16 cases there was gross residual tumor (R2).

The complication rate was 50.33%. A total of 112 patients experienced hypoparathyroidism and 23 patients (7.6%) suffered recurrent laryngeal nerve palsy (17 permanent and 6 transient).

The median postoperative thyroglobulin level was 0.4 ng/mL (range 0.04–5.8), the median stimulated thyroglobulin was 3.4 ng/mL (range 0.04–5000), and the median value on levothyroxine therapy was 0.1 ng/mL (range 0.01–10.4).

A total of 269 patients (89.7%) underwent adjuvant radioiodine therapy, with a mean total radiation dose of 98 ± 19 mCu. 

Of the 300 patients who underwent surgery, 135 (45%) remained at regular follow-up, while the others with incomplete data available were considered lost. In total, 118 (87.4%) were alive, and 17 (12.6%) were deceased.

The mean overall survival at 12 years was 84.8% with a mean of 238 months (Figure 1).

Among the patients with a tumor infiltrating the trachea treated with shaving, three are alive at 33.37 and 101 months, respectively; the others died after 49, 22, and 119 months. One of those with laryngeal infiltration refused laryngectomy and died 25 months later; the other, who submitted to total laryngectomy 5 months after total thyroidectomy, survived for one year.

In total, 107 patients not lost in follow-up had laterocervical lymph node metastases. All of these were functional cervical dissections, including levels II, III, IV, and V. Of these, 11 suffered recurrences, and 86 survived.

Eleven patients (8.1%) experienced relapse after a median of 13 months.

## 4. Discussion

DTC represents a heterogeneous disease, with a very wide range of clinical presentation and prognosis, in most cases it is an indolent tumor and carries an excellent prognosis, with a 10-year overall survival exceeding 90%; therefore, treatment is currently de-escalating [21]. 

Advanced disease is a relatively common event, but in contrast to the localized one, it carries a worse prognosis, requiring an aggressive approach in terms of surgery, indications for RAI, and medical treatment, which leads to an increase in the complication rate [22].

The increasing number of new diagnoses, given the excellent prognosis of most DTCs, has supported the research for new evidence aimed at isolating those clinical and molecular features that can identify the rare cases of aggressive disease, associated with the risk of progression and death, in an effort to implement targeted therapeutic choices and personalized follow-up programs. The goals are saving health care resources where possible and using them where it is most essential.

Many of the clinical and anatomo-pathological features that can be used for this purpose have long been recognized, although for some of them there is not yet a shared opinion in the literature; finally, new acquisitions, also not always shared and reproducible in the different studies, are accumulating in order to identify what the molecular profile of aggressive DTC may be.

In patients with a diagnosis of ADTC, the surgical pathway has to be carefully planned in the preoperative setting, bearing in mind patients’ clinical status, disease extension, and surgical consequences on both the disease course and the quality of life.

Neck dissection, even if limited to the central compartment, puts both parathyroids and recurrent laryngeal nerves at risk. In our series, all patients underwent some form of neck dissection, but even if postoperative complications occurred in more than half of them, in the vast majority, they were transient with spontaneous resolution.

Hypoparathyroidism is the most reported and frightening complication after complete thyroidectomy and can affect 3–49% of patients [23], with an incidence of permanent hypoparathyroidism of 4.11% 6 months after surgery [24]. Low levels of parathyroid hormone (PTH) after surgery and the resulting hypocalcemia may be associated with damage to the parathyroid glands due to an accidental removal of one or more glands or a compromised blood supply due to their dissection [25,26], and this risk may be even higher in patients who are subjected to total thyroidectomy and central neck dissection [25,27,28,29,30].

Among our patients, postoperative hypoparathyroidism had an incidence of up to 37.33%, but of the 112 patients who required calcium and vitamin D supplementation, only 40 (26.49%) still need it.

Our data are consistent with those reported by Pereira in 2005 and, more recently, by Raffaelli and Barczynsky, which confirm that a significantly higher hypoparathyroidism rate is associated to lymphectomy, but, at least in groups of Rome and of Krakow, this increase is due only to transient hypocalcemia [31,32,33].

The use of near-infrared (NIR) autofluorescence of the parathyroid glands and more recently the combination of autofluorescence with fluorescence imaging using indocyanine green (ICG) angiography were proposed as a means of identification and protection of the parathyroid glands during thyroid surgery. It was demonstrated that the preservation of at least one well-vascularized parathyroid gland, as assessed by ICG angiography, predicted the absence of postoperative hypoparathyroidism [34].

In our series, neither autofluorescence nor indocyanine green was used to ease parathyroid visibility and preservation. This is an expanding field, and it may be that more extensive use of these tools will lower the postoperative rates of hypoparathyroidism in the future. 

Addressing the recurrent laryngeal nerve palsies, 23 patients (7.6%) experienced some form of nerve dysfunction. Nerve involvement by the tumor was responsible for 12 out of the 17 instances of permanent damage. In patients in whom intraoperative neuromonitoring showed a loss of signal after the first side, we chose to avoid a staged thyroidectomy because of disease extension.

The recurrent laryngeal nerve is involved in 33–61% of locally invasive thyroid cancers [35], and the decision to sacrifice or preserve a functioning recurrent laryngeal nerve represents a surgical dilemma.

In the literature, there is no consensus about this issue [36,37]. In fact, in 2020, the American Association of Endocrine Surgeons Guidelines for the Definitive Surgical Management of Thyroid Disease in Adults recommended that “Gross thyroid cancer should not be left behind with the intent of preserving parathyroid glands or nerves with the expectation that RAI will kill residual disease, and the risks and benefits of leaving residual tumor to preserve a functioning nerve are considered on a case-by-case basis” [38]. But other scientific societies recommend the shaving of the nerve, leaving no residual gross tumor.

In a recent paper, Lee et al. [39] described for the first time a long follow-up of patients undergone surgery with nerve shaving leaving macroscopic residual disease to preserve functioning recurrent laryngeal nerve encased by a tumor. In their experience, none of the residual tumors showed significant progression to invade the trachea/esophagus or developed anaplastic changes, and only one patient had a significant increase in tumor size and underwent tumor and nerve resection to prevent further extension of the disease.

According to the literature, the treatment of infiltrated RLN is based on the preoperative assessment of vocal function [36,37], the depth of neural invasion, and the presence of non-operatively removable locoregional or remote metastases. It is generally agreed that attempts should be made to preserve the RLN when the preoperative vocal cord function is normal, and intraoperatively, it seems to be a superficial invasion.

When deciding whether to preserve the RLN, the location of RLN invasion should be considered, and resection is warranted if neural invasion occurs near the RLN entrance into the larynx. This is because incomplete tumor resection at this site can lead to cancer progression along the nerve and spread to the larynx [36].

Neoplastic invasion of adjacent structures (airways, esophagus, and large vessels of the neck) may negatively affect the quality of life.

The presence of distant metastases represents the leading cause of cancer-related death.

The use of 18FDG-PET/CT is considered essential in patients with elevated circulating Tg values (generally > 10 ng/mL) with a negative whole-body scan. 18 FDG-PET is most sensitive in patients with an aggressive histologic subtype, including poorly differentiated carcinoma, a high cell variant, and Hurtle cell carcinoma. In patients with Tg < 10 ng/mL after stimulation with recombinant human TSH, the sensitivity of 18FDG is low, ranging from <10% to 30%. Therefore, it is recommended to consider 18FDG-PET only in patients with Tg ≥ 10 ng/mL after stimulation [40].

Many recent data indicate that the aggressive behavior of a DTC (including a high likelihood of tumor recurrence) is more likely when an oncogenic mutation is present, and in particular a BRAF mutation occurring with a mutated promoter of TERT, PIK3CA, TP53, or AKT1 [20].

Treatment of distant metastases is a challenging situation, since it should take into account both the increase in morbidity and mortality rates in metastatic patients, and that individual prognosis is affected by several factors, including primary tumor histology, the spread, the site and the number of metastases (such as brain, bone, lung), age of diagnosis, and RAI avidity.

The prognosis of patients with DTC is generally favorable, even when RAI-responsive metastatic disease is present. For this reason, treatment with I-131 is considered the gold standard in the treatment of metastatic disease.

However, nearly half of the patients with advanced DTC disease become unresponsive or refractory to therapy with I-131, with some of these patients dying within 3 to 5 years, and others also showing long survivorship to a disease that is very slowly progressive.

In these patients, 18 FDG PET plays a double role first in identifying the RAI refractoriness condition and then in the follow-up of patients submitted toTKI therapy to evaluate and monitor the response to treatment [41].

The ATA guidelines describe in detail the sequence in which therapeutic options for metastatic patients should be administered [13].

First of all, potentially curable patients should undergo surgical excision of locoregional metastasis; then, in the case of RAI-responsive disease, 131I therapy is recommended. When metastases are not amenable of previous treatments, ERBT or other focused treatment modalities should be considered.

Asymptomatic patients who present stable disease or slow progression may be treated with thyroid hormone therapy at a TSH-suppressive dose, while those with significantly progressive macroscopic refractory disease should be considered for systematic therapy with kinase inhibitors.

Tyrosine kinase inhibitor therapy should be considered in RAI-refractory patients and in those with rapidly progressive and symptomatic metastatic carcinoma, and/or with imminent danger of disease not otherwise amenable to local control with other approaches. Kinase inhibitors, many of which share the VEGF receptor (VEGFR) as a common target (e.g., sorafenib, pazopanib, sunitinib, lenvatinib, axitinib, cabozantinib, and vandetanib), have recently shown great promise in RAI-refractory metastatic DTC.

In our experience, the extent of treatment has been modulated based on the disease extension and patient status.

This led to encouraging results, with an 84.8% 12-year overall survival with a mean survival of 238 ± 10 months.

Eleven patients experienced disease relapse: seven in the cervical lymph nodes and one in the thyroid bed underwent redo surgery and adjuvant RAI, two in the mediastinum, and one at the laryngeal level.

Among these patients, only one with mediastinal invasion died due to disease progression after 49 months, while the others are alive.

It is essential that these patients are discussed within a multidisciplinary team in order to provide the best treatment in terms of surgery and adjuvant therapies tailored to the patient and disease characteristics.

## 5. Conclusions

Choosing the most suitable therapeutic option for patients with ADTC is difficult because of the many available treatments.

The combination of surgery, even reiterated, RAI, suppressive therapy, and new drugs can provide efficient management of the disease for long periods and, if adequately treated, these patients can achieve prolonged survival, even in the presence of metastasis or disease relapse.

Aggressiveness must be carefully counterbalanced with increased morbidity and a lower quality of life, and the timing of different treatments should also be considered.

According to our experience, it is advisable that ADTC patients be referred to high-volume centers with the availability of an extended multidisciplinary team, consisting of an endocrinologist, an expert endocrine surgeon, an otolaryngologist, a radiologist, a nuclear medicine physician, and an oncologist, in order to receive treatment tailored to both the patient and the clinical and biological features of the neoplasm.

In 2020, the positional statement of the ESES about quality standards in thyroid surgery outlined that there is evidence for a relationship between surgeon volume and outcomes in thyroid surgery with respect to the prevalence of complications.

Then, it gave the definitions of low- and high-volume surgeons as well as high-volume centers based on annual case load and pointed out that since significantly better oncological results in thyroid cancer surgery can be reached if performed by high-volume surgeons, high-volume surgeons should perform thyroid cancer surgery, which is a predictor of the increased risk of surgical morbidity [42].

Future research fields should address how to decrease postoperative morbidity and how to achieve a better quality of life following surgery.

The decision to sacrifice or spare the RLN with the resection of invasive DTC may be guided by an accurate preoperative risk stratification, since BRAFV600E or TERT promoter mutations have been associated with extrathyroidal invasion and an increased risk for disease recurrence, more severe clinicopathological features, and when coexisting the two mutations carry even worse outcomes and higher recurrence rates than any mutation alone [19,20].

Molecular profiling data may help to detect which patients are at a higher risk of developing locoregional relapse when R2 resection is performed (macroscopically incomplete), and therefore need a more aggressive management, and which patients have a lower risk of recurrence and may take advantage of a less invasive treatment.

Moreover, it may be that a more extensive use of the new drugs such as multikinase inhibitors even in the neoadjuvant setting could lead to a de-escalation of surgery in the event of the invasion of adjacent structures and distant metastases.

## Figures and Tables

**Figure 1 jcm-13-00708-f001:**
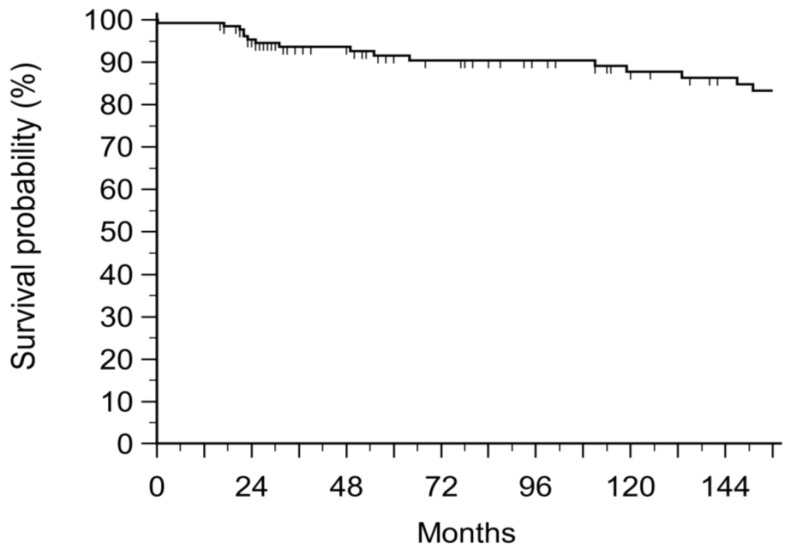
overall survival of 135 patients at regular follow-up.

**Table 1 jcm-13-00708-t001:** Tumor pathology.

	Number	%
TUMOR TYPE		
Papillary	268	89.3
Follicular	32	10.7
LOCAL INVASION	22	
Nerve	12	
Trachea	6	
Esophagus	1	
Larynx	2	
Subcutaneous	1	
NODAL INVASION	260	
Central compartment	121	46.5
Laterocervical	139	53.5
METASTASES	18	
STAGE		
I	227	
II	54	
III	9	
IVA	1	
IVB	9	

**Table 2 jcm-13-00708-t002:** Surgical procedures and complications.

Procedure	N° Patients	%	Complications	N° Patients	%
TT + UCL + BCC	165	55	IPOPTH Tot	112	74.17
TT + BCC	79	26.33	Transient	40	26.49
			Permanent	72	47.68
TT + BCL + BCC	13	4.33	TRLNP	6	3.97
BCL	28	9.33	DRLNP	17	11.25
UCC	11	3.67	SPINAL ACC D	3	1.98
LOB + UCL + BCC	1	0.33	OTHERS	13	8.6
LOB + BCL + BCC	1	0.33			
MET.RES.	2	0.67			
TOTAL	300	100	TOTAL	151	100

TT: total thyroidectomy, UCL: unilateral lateral neck dissection, BCC: central compartment lymphectomy, UCC: unilateral central compartment lymphectomy, BCL: bilateral lateral neck dissection, LOB: lobectomy, MET RES: metastasis resection, IPOPTH: hypoparathyroidism, TRLNP: transient recurrent laryngeal nerve palsy, DRLNP: permanent recurrent laryngeal nerve palsy, SPINAL ACC D: permanent spinal accessory nerve palsy.

## Data Availability

Data of this study are unavailable.

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
