# Peer review of "How to Manage Advanced Differentiated Thyroid Cancer: Step-by-Step Analysis from Two Italian Tertiary Referral Centers"

_jcm, 2024, doi:10.3390/jcm13030708_

Round 1
Reviewer 1 Report
Comments and Suggestions for Authors
Dear Authors,
This study presents a very interesting retrospective work on the experience in treating advanced DTC at two tertiary 22 referral centers in Italy. The authors made a well structured presentation of used method and data presentation, as well as the detailed discussion on each question of importance. I consider your work an interesting contribution of the topic.
Author Response
no change required
Reviewer 2 Report
Comments and Suggestions for Authors
In the article entitled “How to Manage Advanced Differentiated Thyroid Cancer: 2 Step-By-Step Analysis from Two Italian Tertiary Referral Centers” by Sartori et al., the authors analyze their experience about the treatment of advanced DTC in 2 tertiary surgical centers. They included 300 patients.
Total thyroidectomy with lymph node dissection of variable extent was performed in 257 patients and lobectomy only in 2 patients because of previous surgery.
269 patients had radioiodine therapy.
Regular follow-up was done in 135 patients and data were available. Of these only 118 were alive. The authors reported overall survival at 12 years of 84.8% and the recurrence in 8.1% of patients after 13 months.
Reported rate of definitive hypoparathyroidism was 26.4%.
They reported permanent laryngeal nerve palsy in 12 patients because of infiltrated by cancer. In 8 patients the nerve was sacrificed.
2 patients had laryngeal infiltration and in 6 patients they found tracheal involvements that was treated with shaving resection.
This article is very interesting in that the authors emphasize the importance of aggressive treatment of advanced cancers at a time when there is a tendency to therapeutic deescalating.
The other strong point of this manuscript is the importance of taking into account the increased risk of complications in taking care of advanced cancer.
For example, about the infiltration of the recurrent nerve, since there is no consensus on the matter, the authors address the dilemma of whether to resect or preserve the nerve in case of infiltration. It should be kept in mind that in most cases patients respond well to iodine therapy except in the iodine-resistant forms. Therefore, in most cases it is possible to preserve the nerve. This detail could also be added.
I believe that in order to improve the quality of manuscript, the authors could answer and clarify the following point.
-In Table 1, the authors describe the details of the surgery performed, but these data are partially included in the text. It is recommended that these data also be included in the results section.
Similarly, they mentioned complications in Table 1 but not in the results, although they mention them in the discussion. I advise them to add such data in the results as well.
-Tab 1 shows the acronym “IPOPTHT” is which refers to transient hypoparathyroidism. Wouldn't it be better to use THPT?
-Similarly, for RLNPT and RLNPD acronyms, it would be recommended to use TRLNP and DRLNP.
-The authors stress the concept of the risks of aggressive surgery. However, numerous publications argue that these risks, when the surgery is performed, by an experienced high-volume team, are reduced. Therefore, I invite them to read about the risk of hypoparathyroidism the recent following article:
Langenbecks Arch Surg. 2023 Oct 10;408(1):393. doi: 10.1007/s00423-023-03130-w. Accidental parathyroidectomy during total thyroidectomy and hypoparathyroidism in a large series of 766 patients: incidence and consequences in a referral center.
-The authors should review acronyms included in the text and clarify them when first used.
The following are a few examples:
-Line 41: PTC
-Line 59: SNC
-Line 83: DTC
-Line 105: AUS
-Line 167: tki
-Line 57: “carcinoma metastasizes by the hematogenous route.
Please replace “hematogenous route” with “hematogenous spread”.
-Line 139-40: “extended thyroidectomy (total thyroidectomy plus resection of adjacent structures involved by tumor”.
The authors should specify what they mean by resection of adjacent structures involved.
-Line 141-143 “unilateral central compartment node clearance”
It would be more appropriate to replace "clearance" with dissection.
-Line 156: punctuation is missing
-Line 242: In the sentence there is punctuation to be removed.
-Line 261-263: The authors repeated the sentence twice. The same phrase is given in line 244-246. Please correct.
-Line 264-270: It is recommended to rephrase this period because it is not very clear.
-301-310: The authors report part of the results in this section. It would be better to report these data in the results section.
-369: In the sentence there is punctuation to be removed.
Reviewer 3 Report
Comments and Suggestions for Authors
- In your study, the definition of advanced disease includes nodal invasion, involvement of adjacent structures, or distant metastasis. Please provide a clearer address of this definition, specifying the number of patients falling into each criterion. For nodal invasion, clarify whether it refers to the central or lateral compartment. For adjacent structures, please specify the location (trachea, esophagus, recurrent laryngeal nerve, carotid artery, etc.) and the number of patients in each category, along with the metastatic site and corresponding patient numbers. While recognizing the absence of a widely accepted definition for advanced thyroid cancer, the data provided is too limited and insufficient to convincingly portray the severity of the enrolled patients' diseases.
- In the materials and methods, you mention the operation type, including extended thyroidectomy (total thyroidectomy plus resection of adjacent structures involved by the tumor). However, this procedure does not appear in Table 1. Can you provide information on how many patients had resection of adjacent structures (trachea, n=; esophagus, n=; nerve=;)? Specify the type of resection (tumor shaving, structure resection?). The classification of residual tumor at the primary site after treatment (R0 resection, R1 resection, or R2) is crucial for predicting prognosis.
- The abbreviations used in Table 1 are uncommon and difficult to understand, and the authors did not show the full name of CCL, UCC. The extent of lymph node dissection should be specified, mentioning whether it includes the central compartment and/or lateral compartment.
- The percentage of complications presented in Table 1 is confusing. It seems you have 151 complications, and 47.68% (72/151) is transient hypoparathyroidism, 26.49%(40/151)is permanent. However, the usual definition of the complication rate is the number of affected patients/total patients.
In lines 244-246, you mention postoperative hypoparathyroidism is 37.33% (112/300?), and only 40 (26.49%) still need it. 40/112 = 35.7%, and the 26.49% is equal to 40/151 (Table 1). It appears that there might be a mix-up in the two definitions of the percentage of complications. Please check the numbers for accuracy.
- Please provide the basic patient characteristics, including the patient number in each cancer histology (papillary, follicular, Hurthle cell), TNM stage, and thyroglobulin levels (postoperative, TSH stimulated, TSH suppressed).
Round 2
Reviewer 3 Report
Comments and Suggestions for Authors
I believe the following point requires further clarification.
Table 2 requires clarification regarding the extent of lymph node dissection. The terminology used is quite confusing.
- Since UCC stands for unilateral central compartment lymphectomy, I presume CCL (central compartment lymphectomy) refers to bilateral central compartment lymphectomy. Please confirm this and consider using BCC as an abbreviation, which would be more consistent with the abbreviation rule applied in UCL and BCL.
- For UCL (unilateral cervical lymphectomy) and BCL (bilateral cervical lymphectomy), please specify the extent of lymph node dissection for the term "cervical lymphectomy." This term is currently ambiguous, and clarification is needed. It is suggested to explicitly state whether it represents lateral neck dissection (level II, III, IV, V).
